# MAC-ResNet: Knowledge Distillation Based Lightweight Multiscale-Attention-Crop-ResNet for Eyelid Tumors Detection and Classification

**DOI:** 10.3390/jpm13010089

**Published:** 2022-12-29

**Authors:** Xingru Huang, Chunlei Yao, Feng Xu, Lingxiao Chen, Huaqiong Wang, Xiaodiao Chen, Juan Ye, Yaqi Wang

**Affiliations:** 1College of Media Engineering, Communication University of Zhejiang, Hangzhou 310042, China; 2School of Electronic Engineering and Computer Science, Queen Mary University of London, Mile End Road, London E1 4NS, UK; 3Department of Ophthalmology, The Second Affiliated Hospital of Zhejiang University School of Medicine, Hangzhou 310009, China; 4School of Computer Science and Technology, Hangzhou Dianzi University, Hangzhou 310005, China

**Keywords:** deep learning, eyelid tumor classification, digital pathology images, MAC-ResNet

## Abstract

Eyelid tumors are tumors that occur in the eye and its appendages, affecting vision and appearance, causing blindness and disability, and some having a high lethality rate. Pathological images of eyelid tumors are characterized by large pixels, multiple scales, and similar features. Solving the problem of difficult and time-consuming fine-grained classification of pathological images is important to improve the efficiency and quality of pathological diagnosis. The morphology of Basal Cell Carcinoma (BCC), Meibomian Gland Carcinoma (MGC), and Cutaneous Melanoma (CM) in eyelid tumors are very similar, and it is easy to be misdiagnosed among each category. In addition, the diseased area, which is decisive for the diagnosis of the disease, usually occupies only a relatively minor portion of the entire pathology section, and screening the area of interest is a tedious and time-consuming task. In this paper, deep learning techniques to investigate the pathological images of eyelid tumors. Inspired by the knowledge distillation process, we propose the Multiscale-Attention-Crop-ResNet (MAC-ResNet) network model to achieve the automatic classification of three malignant tumors and the automatic localization of whole slide imaging (WSI) lesion regions using U-Net. The final accuracy rates of the three classification problems of eyelid tumors on MAC-ResNet were 96.8%, 94.6%, and 90.8%, respectively.

## 1. Introduction

Eyelid tumors are complicated and diverse, including tumors of the eyelid, conjunctiva, various layers of ocular tissues (cornea, sclera, uvea, and retina), and ocular appendages (lacrimal apparatus, orbit, and periorbital tissues) [1,2,3]. Primary malignant tumors of the eye can spread to the periorbital area, intracranially or metastasize systematically, and malignant tumors of other organs and tissues throughout the body can also metastasize to the eye. Therefore, eyelid tumors cover almost all histological types of tumors in the whole body and are widely representative, which can be the best thing object of study for pathological diagnosis of tumors.

Basal cell carcinoma (BCC) is a type of skin cancer that originates in the basal cells of the epidermis. It is the most common type of skin cancer, often occurring on sun-exposed areas of the body. Meibomian gland carcinoma (MGC) is a rare form of cancer affecting the meibomian glands in the eyelid, which secrete an oily substance for eye lubrication. MGC typically presents as a slow-growing lump on the eyelid, potentially mistaken for a benign cyst. Cutaneous melanoma (CM) is a type of skin cancer arising from pigment-producing cells known as melanocytes. It is less common than BCC, but more aggressive and capable of spreading to other parts of the body if left untreated. CM typically appears as a dark-colored new or changing mole or patch of skin, but may also present as a pink or red patch. According to morbidity studies, BCC is the most common malignant eyelid tumor, followed by CM and MGC [4,5,6,7].

Computed Tomography (CT) and Magnetic Resonance Imaging (MRI) have their limitations that affect their respective clinical applications. A biopsy is an important tool for physicians to finally diagnose eyelid tumors, and pathological diagnosis is the “gold standard” of diagnosis, and observation and analysis of histopathological images of biopsies is an important basis for physicians to formulate the best treatment plan [8,9,10]. The observation and analysis generally require qualitative, localization, and scoping judgments. However, the extreme shortage of human resources and overload in pathology departments are far from meeting the needs of clinical patients for accurate and efficient diagnostic pathology. Accurate diagnosis of BCC, MGC, and CM is essential for optimal patient outcomes, as early diagnosis is a key factor in determining the likelihood of a cure. While the physical appearance of these skin cancers may be distinctive, a biopsy is typically required for definitive diagnosis. Histologically, these types of eyelid tumors can be similar, making it possible to misdiagnose based on histological slides alone. The importance of accurate diagnosis cannot be overstated, as 90% of patient survival has been associated with early detection in cases where pathology-based diagnosis is involved.

Inspired by the concept of knowledge distillation [11], we have trained a teacher-student model to classify and segment eyelid tumors with good performance and a smaller, more efficient student network. In this paper, we will study the classification and segmentation of tumors based on meaningful learning methods using eyelid tumor pathology images, and the overall flowchart of the network is shown in Figure 1. The main contribution of this paper includes the following points:(1)Propose the network model called Multiscale-Attention-Crop-ResNet (MAC-ResNet). This network model can achieve 96.8%, 94.6%, and 90.8% accuracy in automatically classifying three ocular malignancies, respectively.(2)By training the student network ResNet with MAC-ResNet as the teacher network with the help of the knowledge distillation method, we made the smaller-scale network model to obtain better classification results on the eyelid tumor dataset, which called ZLet dataset.(3)We train three targeted segmentation networks for each of the three different malignant tumors, which enable us to segment the corresponding tumor locations well. With the help of the classification and segmentation networks, we diagnose the disease and the rapid localization of the lesion area.

## 2. Related Work

The pathology segmentation and classification of eyelid tumors is a crucial aspect of ocular oncology as early diagnosis and treatment can significantly improve patient outcomes. One of the most common types of skin cancer that can occur on the eyelid is basal cell carcinoma (BCC). This type of cancer arises from the basal cells in the skin and is often caused by prolonged exposure to ultraviolet radiation. While BCC is not typically life-threatening, if left untreated it can cause significant damage to the skin and surrounding tissues. Cutaneous melanoma (CM), on the other hand, is a more aggressive form of skin cancer that originates from the pigment-producing cells in the skin. While less common than BCC, it has a higher likelihood of spreading to other parts of the body and can be deadly if not caught early. A rare type of cancer that can affect the eyelid is meibomian gland carcinoma (MGC), which arises from the meibomian glands that produce oil to keep the eye moist. MGC is generally more aggressive than BCC and can spread to other parts of the body if not treated promptly.

Accurately distinguishing between these three types of tumors is vital for treatment planning and research. Patients diagnosed with BCC may be treated with surgical or other local interventions to remove the tumor, while those diagnosed with cutaneous melanoma may require more aggressive treatment approaches, such as surgery, radiation therapy, or chemotherapy, in order to prevent the spread of the cancer. In addition, accurate classification and segmentation of eyelid tumors has significant value for research, including the study of the biology and genetics of these tumors, the evaluation of treatment response and disease progression, and the development of diagnostic and treatment algorithms. Therefore, a reliable method for classifying and segmenting eyelid tumors is necessary.

In recent years, with the development of deep learning in the field of computer vision, the study of medical image processing based on deep learning has become a popular research topic in the field of computer-aided diagnosis [12,13,14], and methods using deep learning are gradually being used for the diagnosis and screening of a variety of ophthalmic diseases, however, less research has been conducted on eyelid tumors.

In 2019, Hekler et al. will use a pre-trained ResNet50 [15] Network for training 695 whole slide images (WSIs) by migration learning to reduce the diagnosis error of benign moles and malignant melanoma [16]. Xie et al. used the VGG19 [17] network and ResNet50 network to classify patches generated from histopathological images [18]. In 2022 Wei-Wen Hsu et al. proposed CNN for the classification of glioma subtypes using mixed data of WSIs and mpMRIs under weakly supervised learning [19], Nancy et al. proposed DenseNet-II [20] model through HAM10000 data set and various deep learning models to improve the accuracy of melanoma detection. At the 2018 ICCV conference, Chan et al. proposed the HistoSegNet method for semantic segmentation of tissue types, using an annotated digital pathology atlas (ADP) for patched training, computation of gradient-weighted class activation maps, which outperforms other more complex weakly supervised semantic segmentation methods [21]. X Wang et al. based on the idea of model integration designed two complementary models based on SKM and scSEM to extract features from different spaces and scales, the method can directly segment the patches of digital pathology images pixel by pixel and no longer depends on the classification model [22].

Although computer vision has made some progress in the field of tumor segmentation, automated analysis studies based on eyelid tumor pathology are very rare due to the lack of dataset. In 2018, Ding et al. designed a study using CNN for the binary classification of malignant melanoma (MM) and the whole slide image-level classification was realized using a random forest classifier to assist pathologists in diagnosis [23]. In 2020, Wang et al. trained CNN on patch-level classification and used malignant probability to embed patches into each WSI to generate visualized heatmaps and also established a random forest model to establish WSI-level diagnosis [24]. Y Luo et al. performed patch prediction by a network model based on the DenseNet-161 architecture and WSI differentiation by an integration module based on the average probability strategy to differentiate between eyelid BCC and sebaceous carcinoma (SC) [25]. Parajuli et al. proposed a novel fully automated framework, including the use of DeeplabV3 for WSIs segmentation and the use of pre-trained VGG16 model, among others, to identify melanocytes and keratinocytes and support the diagnosis of melanoma [26]. Ye et al. first proposed a Cascade network to use the features from both histologic pattern and cellular atypia in a holistic pattern to detect and recognize malignant tumors in pathological slices of eyelid tumors with high accuracy [27]. Most of the above studies are based on existing methods and do not make significant modifications to the segmentation network. Some studies only focus on the recognition task and assist doctors in the diagnosis through classification, without involving tumor region segmentation due to the lack of a large-scale segmentation dataset in this task. Segmentation task is an important factor in evaluating the tumor stage and is also the basis for quantitative analysis. Our proposed method is able to simultaneously perform eyelid tumor classification and segmentation tasks based on histology slides through the design of the network architecture.

There are various factors that can increase the complexity of segmenting BCC, CM, and MGC in histology slides. The subtle differences in appearance that these tumors may exhibit compared to normal tissue, which can make them difficult to distinguish. Additionally, early-stage cancers may be more challenging to detect due to their small size and potential lack of discernible differences from normal tissue. To address these issues, we proposed the MAC-ResNet based on the teacher-student model for accurate classification and segmentation of eyelid tumors.

The teacher-student model is a machine learning paradigm in which a model, referred to as the “teacher”, is trained to solve a task and then another model, referred to as the “student”, is trained to mimic the teacher’s behavior and solve the same task. The student model is typically trained on a smaller dataset and with fewer resources (e.g., fewer parameters or lower computational power) than the teacher, with the goal of achieving similar or improved performance at a lower cost.

The teacher-student model is also known as the knowledge distillation or model compression approach. It is often used to improve the efficiency and performance of machine learning models, particularly when deploying them in resource-constrained environments such as mobile devices or Internet of Things (IoT) devices. In the teacher-student model, the teacher model is first trained on a large dataset and then used to generate “soft” or “distilled” labels for the student model, which are more informative than the one-hot labels typically used for training. The student model is then trained using these soft labels and the original dataset, with the goal of learning to mimic the teacher’s behavior. There are several variations of the teacher-student model, which can be divided into logits method distillation and feature distillation based on the transfer method. In this study, we adopt the logits method distillation. The concept of knowledge distillation and teacher-student model first appeared in “Distilling the knowledge in a neural network” by Hinton et al., and was used in image classification. Later, knowledge distillation was widely used in various fields of computer vision, such as face recognition [28], image/video segmentation [29], etc. In addition, it has also been applied in natural language processing (NLP) fields such as text generation [30], question answering systems [31], and others. Furthermore, it has also been applied in areas such as speech recognition [32] and recommender systems [33]. Finally, knowledge distillation has also been widely used in medical image processing. Qin et al. proposed a new knowledge distillation architecture in [34], achieving an improvement of 32.6% on the student network. Thi Kieu Khanh Ho et al. proposed a self-training KD framework in [35], achieving student network AUC improvements of up to 6.39%. However, this is the first time that knowledge distillation has been used in the classification of dermatopathology images.

## 3. Methods

First, we normalize and standardize the input data features and use a random combination image processing method to perform image expansion and enhancement. Then, we newly proposed a network structure (MAC-ResNet) that performs well on the classification task on the ZLet dataset, but the whole model structure is complex, consumes a lot of computational resources throughout the training process, and the speed of algorithm inference is slow. Therefore, we adopt the model compression method of knowledge distillation, use MAC-ResNet as the teacher network and ResNet50 as the student network, and achieve good results of the small volume student network ResNet50 in the classification of digital pathological pictures of eyelid tumors by using the knowledge of the teacher network to guide the training of the student network. Thus, this paper achieves automatic classification of three types of malignant tumors and enables automatic localization of lesion areas using U-Net [36].

### 3.1. MAC-ResNet

To solve the problem of low accuracy of fine-grained classification, we first propose the Watching-Smaller-Attention-Crop-ResNet (WSAC-ResNet) structure. It combines the Backbone-Attention-Crop-Model (BACM) module, the residual nested structure Double-Attention-Res-block, the SPP-block module, and the SampleInput module.

For the fine-grained classification problem, this paper refers to the fine-grained classification model WSDAN [37] and modifies it to design the Backbone-Attention-Crop-Model (BACM) module. From Figure 2, we can learn that the BACM Model consists of three parts. They are the backbone network, the attention module [38], and the AttentionPicture generated by cropping the original image according to the AttentionMap.

We crop and upsampling key regions of the images to a certain size according to the attention parameters, aiming to guide data for enhancement through the attention mechanism. Before the Feature Map of the neural network is input to the fully connected layer, it is input to the Attention model, and X Attention maps are obtained by convolution, dimensionality reduction, and other operations, each Attention map represents a feature in the picture, and one Attention map is randomly selected among the X Attention maps Then the normalization operation is performed on the Attention map. The normalization operation is as (1).
(1)$$A_{k}^{*}=\left(A_{k}-\min\left(A_{k}\right)\right)/\left(\max\left(A_{k}\right)-\min\left(A_{k}\right)\right)$$

The value of the newly obtained Attention map is changed to 1 for elements with values more significant than the threshold $\theta_{c}$ and set to 0 for elements at other locations to generate a mask of locations worthy of strategic attention. The original image is cropped according to the generated mask against the original image to get the image of important regions and upsampling to a certain size, and then re-input into the neural network after data enhancement processing. When calculating the loss of the network model, the mean of the predicted and labeled loss of the original image and the predicted and labeled loss after cropping and re-inputting into the model is seen as the ultimate loss.

The backbone network is a neural network based on ResNet50 with a modified input structure named SampleInput, specifically by replacing a 7*7 convolutional layer with three 3*3 convolutional layers to enhance the network depth and ensure they have the same perceptual field; the network uses a double-layer nested residual structure Double-Attention-Res-block (DARes-block), which can fuse the deep layer with the shallow layer and the feature maps of the middle layer; SPP-block, which originated from SPPNet [39], is used to solve the training problem for different image sizes.

To further improve the classification of the network, the loss function and the learning rate adjustment strategy of this network will be optimized.

For the classification of unbalanced samples, the focal loss function [40] is used, which is a modification of the cross-entropy loss function, as (2).
(2)$$FL\left(p_{t}\right)=-a_{t}{\left(1-p_{t}\right)}^{\gamma }\log\left(p_{t}\right)$$

We use CosineAnnealingLR [41] to adjust the learning rate. It is used to change the magnitude of the learning rate by the cosine function, and each time the minimum point is reached. The next step resets the value of the learning rate to the maximum value to start a new round of decay.

We named the network that uses the above modules and policies as Multiscale-Attention-Crop-ResNet (MAC-ResNet).

### 3.2. Network Optimization Based on Knowledge Distillation

First, the teacher network with a complex model and good performance is trained, then the trained teacher network guides the training of the student network, and the trained student network is used to classify the dataset [42]. The main principle of the teacher network guiding the training of the student network is that the soft labels output by the teacher network and the output of the soft label by the student network are combined to coach the student network to complete the training of the hard labels (as shown in Figure 3). Soft labeling means that the predicted output of the network is divided by the temperature coefficient T and then the softmax operation is performed, which makes the result values between 0 and 1 with a more moderate distribution of values, while hard labeling means that the predicted output of the network is directly softmaxed without dividing by T [43].

Traditional segmentation networks consume a large amount of computing resources during the entire training process and has a slow inference speed during the training of large pathology dataset. It is possible to compress the segmentation model to generate a smaller network with similar performance. We adopt the model compression method of knowledge distillation, using the aforementioned MAC-ResNet as the teacher network. Then, we use the simple and classic ResNet50 as the student network. Finally, we achieve good classification results on the ocular tumor pathology image dataset using the relatively simple student network. Knowledge distillation is a method proposed by Hinton et al. [42], in which a complex and large model is used as the Teacher model, while the student model has a simpler structure. The Teacher model assists in the training of the student model, which has weaker learning ability, by transferring the knowledge it has learned to the student model, thereby enhancing the Student model’s generalization ability. Therefore, in the knowledge distillation process, the teacher network is usually a network with a complex structure, slow inference process, high consumption of computer resources, and good model performance, while the student network is usually a network with a simpler structure, fewer parameters, and poorer model performance. The process of using knowledge distillation is as follows: first, we train the complex and well-performing teacher network (MAC-ResNet), then guide the training of the student network (ResNet50) using the trained teacher network, and finally use the trained student network to classify the dataset. The teacher network guides the training of the student network by providing the student network with soft labels, or the probabilities of each class predicted by the teacher network, instead of hard labels (as shown in Figure 3), which is the one-hot encoded labels of each class. For soft labeling, the predicted output of the network is divided by the temperature coefficient T and then the softmax operation is performed, which makes the result values between 0 and 1 with a more moderate distribution of values, while hard labeling means that the predicted output of the network is directly softmaxed without dividing by T [43]. This helps the student network learn from the rich information provided by the teacher network. The softmax process can be denote as:(3)$$\mathrm{Softmax}\left(T\right)=\left(\exp\left(z_{i}/T\right)\right)/\left(\sum_{j}\exp\left(z_{i}/T\right)\right)$$

The loss of the MAC-ResNet network consists of two parts, which are the loss between the predicted value and the label of the first original input picture and the loss between the predicted value and the label of the network model after the attention-guided cropping to generate AttentionPicture into the network, and the weighted sum between them is the final loss. The proposed loss function of the whole training process after using MAC-ResNet as the teacher network and ResNet50 as the student network is shown in (4) and (5).
(4)$$\begin{array}{cc} L_{KD}= & \left(1-a\right)L_{f}\left(\;S{\;}_{HP},\;\mathrm{label}\;\right)+\left(\;a\;*T*T\right)L_{1}\left(\;S{\;}_{SP},\;T{\;}_{SL}\right) \end{array}$$
(5)$$\begin{array}{cc} T_{loss}= & 1/2*L_{KD}+1/2*L_{f}\left(\;T{\;}_{AHP},\;\mathrm{label}\;\right) \end{array}$$
where $S_{SP}$ refers to the output of the hard label by the student network, $S_{SP}$ refers to the output of the soft label by the student network, $T_{SL}$ refers to the soft labels generated by the teacher network for the original picture prediction, and $T_{AHP}$ refers to the hard labels predicted by the teacher network based on the AttentionPicture (the labels are softened only for the results of the original picture prediction). Besides,$L_{KD}$ refers to the loss of Knowledge Distillation, and $T_{loss}$ refers to the total loss. $L_{1}$ is the $K_{L}$ scattering loss function (Kullback-Leibler divergence), $L_{f}$ is the focal loss function. T is the temperature coefficient, the larger the temperature coefficient, the more uniform the output data distribution.

After using knowledge distillation, the lightweight network model ResNet50, which is a student network, showed a significant improvement in the classification of the ZLet dataset.

## 4. Experiment and Result

### 4.1. Data and Process

#### 4.1.1. Data Gathering

We collected an eyelid tumor segmentation dataset, ZJU-LS eyelid tumor (ZLet) dataset, including 728 whole slide images and corresponding tumor masks. This is the largest eyelid tumor dataset ever reported. Over a period of seven years from January 2014 to January 2021, we collected pathological tissue slides from 132 patients treated at the Second Affiliated Hospital, Zhejiang University School of Medicine (ZJU-2) and Lishui Municipal Central Hospital (Lishui). We then used hematoxylin and eosin (H&E) staining to visualize the components and general morphological features of the tissue slides, enabling pathologists to observe and annotate them. Finally, we used KF-PRO-005 (KFBio, Zhejiang, China) to digitally amplify all pathological tissue slides at 20× magnification, resulting in a total of 728 whole slide images, including 136 BCC, 111 MGC, and 481 CM, as shown in the figure. These fully-annotated WSI were observed, diagnosed, and labeled by three experienced pathologists (>5000 h experience). The area marked by the doctors only contained the tumor of that category. To facilitate deep learning, we divided these WSI into training, validation, and testing sets. The training set included 425 CM, 124 BCC, and 81 MGC. The validation set included 48 CM, 12 BCC, and 9 MGC. The testing set included 8 CM, 21 BCC, and 21 MGC. Some examples are demonstrate in Figure 4.

#### 4.1.2. Data Preprocessing

During training, to decrease the need for memory and speed up the training process, we divided the full-field digital slices into small blocks based on the diseased regions labeled by the physician and then cropped the diseased regions. When generating the patch, the mask image is compared with the pathology image, and we crop the pathology image area corresponding to the white area of the mask image, the size of the crop is 512×512, stride = 256, which means there is an overlap of the image to crop. If the diseased area(the area with the value of 1 in the mask) in the current clipping area is more than 3/4 of the total area, the patch is kept, otherwise, it is discarded. The purpose of this is to prevent the patch from containing only a small number of diseased regions. After obtaining all the tiny patches, the cropped data were cleaned of images smaller than 330 kb, because images smaller than 330 kb in size contain a small number of scattered tissue regions, and such images can interfere with the training of the neural network. We also normalized and standardized the data features before feeding them into the neural network.

#### 4.1.3. Data Augmentation

Before a batch of images is input to the neural network, we randomly select random flips, random rotations, horizontal flips, and vertical flips, modify the saturation of the image, add Gaussian noise, extract the outline of the image, and finally apply a combination of smoothing operations on the image, so that the same image in different training batches will generate many different transformed images, and the operation can be performed faster to get The enhanced results are obtained faster and do not require additional storage space to store the images. This operation not only enriches the data input to the neural network but also increases the features of the data, allowing the neural network to learn more features and enhancing the generalization ability of the model.

### 4.2. Ablation Study

To explore the effect of having the nested residual module DARes-block in different positions in ResNet50 on the experimental results, we designed an experiment keeping the original input unchanged and using the network structure of ResNet50+BACM. Table 1 shows the best results on the validation set for each group of experiments using DARes-block for the network structure in this chapter, where ACC denotes accuracy, Spec denotes specificity, Recall denotes recall, and 0-ACC is the accuracy of class 0.

From Table 1, we can see that the use of DARes-Block improves t network performance whether the modified residual structure block DARes-Block is used in layer2, layer3, and layer4 separately or in the combination of layer2, layer3, and layer4. One of the best experimental results is the experiment using DARes-Block in both layer2 and layer4 structures. Through our analysis, we determine that it is because using the DARes-Block structure in layer4 enables us to obtain more detailed features. These features were then fed directly into the attention mechanism without going through other convolutional or pooling layers. Although an experiment used the DARes-Block structure in each layer, the experimental results were not the best because using the structure in each layer increases the complexity of the model and is prone to overfitting problems, resulting in poor test results. At the same time, we also found that the accuracy of all four categories improved after using the DARes-Block structure, and it is no longer obvious to focus on one category, which indicates that the structure is effective in fine-grained classification.

The next step is to explore the effect of modifying the input module on the network performance based on the addition of the residual structure block DARes-Block network structure at layer2+layer4.

Table 2 shows the experimental results for the network with or without modifying the input module, which are the best results for each group of experiments on the validation set.

The accuracy of the test set was improved somewhat by modifying the input structure on the ResNet50+BACM+DARes-Block network model structure from Table 2 above, but the improvement was not too significant. Since the modification of the input module did not cause a considerable increase in network complexity and did not additionally increase the training time of the network, we kept the modified input module.

Then, to investigate the role of SPP-Block, we designed the experiments still using the control variable method. The difference between the experiments is whether the SPP-Block module is used; both are network structures with modified input structures on the ResNet50+BACM+DARes-Block model structure, except for this difference.

From the comparison experiments in Table 3, we can see that the model’s performance is slightly improved with the SPP-block than without this structure, indicating that the SPP-block is beneficial for improving the model’s performance. Ultimately, we refer to the structure using the above modification as WSAC-ResNet.

To verify the effect of different loss functions on the classification effect of WSAC-ResNet, we designed experiments to compare the effect of three different loss functions on the classification effect. The experiments were conducted using the control variables method, and the three sets were identical except for the loss function, which used the same WSAC-ResNet network structure and parameters as the previous experiments. When using Focal loss, the values of and are set as default values, and the smoothing factor is set to 0.1, and the results of the experiment with the best effect on the validation set are also taken in the comparison experiment.

Figure 5 shows the comparison of the loss of the WSAC-ResNet network model during training using different loss functions. Among them, Labelsmoothing mitigates the overfitting problem by a regularization method that adds noise and reduces the weight of the true label in calculating the loss [44]. The loss values are recorded at 2 intervals during training. In this iterative analysis of the training losses for focal loss, cross-entropy loss, and label loss, it was observed that the focal loss initially started at a value of 1.8, but quickly dropped to 1.0 after about 200 steps. The cross-entropy loss, on the other hand, remained relatively stable at a value of around 1.4, while the label loss was the highest of the three losses at the beginning, but dropped to the middle of the other two losses after about 200 steps. After 500 steps, the three losses seemed to stabilize, with focal loss remaining at 0.66, label loss stable at 0.74, and cross-entropy loss staying at 0.85. This pattern of loss values suggests that the model may be more sensitive to the focal loss and may be learning more effectively using this loss function compared to the other two loss functions. Due to the uneven distribution of data among different classes and the inability of the labeled tags to be totally accurate, focal loss has an advantage both in terms of training time and performance.

From the comparison in Table 4, we know that using focal loss is better than using Cross Entropy, label smoothing model, where the accuracy is improved by about 2% when using focal loss, and the accuracy of class 1 disease and class 2 disease is improved from 0.7858 and 0.826 to 0.862 and 0.871. The accuracy is improved by about 1% when using label smoothing. Observing the loss comparison graph, at the late loss convergence, the loss using focal loss is lower than that using Cross Entropy, and the label smoothing loss converges slower and at the early stage, the value of his loss is larger than that of both other approaches.

Therefore, both focal loss and label smoothing have improved the classification effect of WSAC-ResNet, but the WSAC-ResNet network model combined with the focal loss function is more effective because focal loss can alleviate the problem that the network model focuses on training a certain class due to the imbalance of dataset between classes. Therefore, the loss function of WSAC-ResNet is set to focal loss.

From Table 5 above and the previous experimental results, we can learn that the classification accuracy of the two strategies, focal loss and CosineAnnealingLR, when used in combination, reached 0.9023, naming the network model with the combination of WSAC-ResNet, focal loss, and CosineAnnealingLR as Multiscale- Attention-Crop-ResNet (MAC-ResNet).

### 4.3. Preformance

#### 4.3.1. Network Training

Our experiments are based on the Pytorch. The experimental operating system is Ubuntu 20.04, With AMD R9 5950X, two NVIDIA 3080 10 GB graphic cards, and 128 GB of RAM. We trained our network from scratch for 50 epochs. The batch size is set to 8 for all experiments and a learning rate of 0.00001.

#### 4.3.2. Evaluation Metrics

To evaluate the classification performance of our network, we used various evaluation metrics including Sensitivity, Specificity, and Accuracy. Also, we used two evaluation metrics, IOU and Dice, to evaluate the segmentation performance of our network. Their formulas are as follows:(6)$$\;\mathrm{Sensitivity}\;=\frac{TP}{TP+FN}$$
(7)$$\;\mathrm{Specificity}\;=\frac{TN}{TN+FP}$$
(8)$$\;\mathrm{Accuracy}\;=\frac{TP+TN}{TP+TN+FP+FN}$$
(9)$$\;\mathrm{IOU}\;=\frac{TP}{FN+TP+FP}$$
(10)$$\;\mathrm{Dice}\;=\frac{2\times TP}{FN+TP+TP+FP}$$

#### 4.3.3. Patch-Level Classification

To demonstrate the performance of our model for the three eyelid tumor classification problems, we used the classical metrics sensitivity, specificity, and accuracy in the classification problem to measure the classification results. As shown in Table 6, the classification results for all three eyelid tumors are relatively high, which reflects the significant effectiveness of our model in the triple classification problem of eyelid tumors.

#### 4.3.4. WSI-Level Results

At the WSI level, we segmented the classified and reorganized WSI map and the original WSI map with the traditional U-Net, and the results were combined to finally segment the focal regions of the three eyelid tumors. The segmentation results are shown in Table 7, and their metrics indicate that their method can meet the need for rapid determination of the lesion regions, and the segmented images are visualized as shown in the ground truth in Figure 6.

This segmentation result can suggest that the doctor should focus on this region, which has a high probability of having some kind of tumor and provide aid to the doctor to diagnose which kind of tumor the pathological image contains and where the tumor is located, which can help to remove the tumor later. In addition, the classification results on the patches can be combined to form an attention map, and by processing the attention map, we can get the feature maps of the model for the normal and tumor regions (as shown in Figure 6 the attention, feature map). These tumor feature maps can further help doctors to analyze the tumor in pathology slides, which is a reliable basis for doctors’ diagnostic analysis.

## 5. Conclusions

The segmentation based on pathology slides is usually time consuming. In order to improve efficiency, we have adopted the knowledge distillation method, inspired by Hinton et al., to train a student network using a MAC-ResNet as the teacher network, enabling the student network to achieve good accuracy on the target task even with a small capacity. In addition, by using U-Net to achieve automatic localization of the lesion area, we can provide a reliable foundation for the diagnosis of pathologists and improve the efficiency and accuracy of diagnosis. We have applied this method to pathology tumor detection for the first time and have successfully verified the practicality of the teacher-student model in the field of pathology image analysis. Finally, the accuracy of MAC-ResNet on the three target tasks was 96.8%, 94.6%, and 90.8%, respectively. However, this study also had some regrets that we were not able to conduct extensive experiments on this data to widely verify the performance of different methods under the teacher-student framework. Another limitation of this study is that it only studied BCC, MGC, and CM, while eyelid tumors include other diseases, so more data sets will be needed in the future. We are currently working on a larger data set, ZLet-large, based on ZLet. ZLet-large includes over a thousand eyelid tumor pathology images and an increased number of disease types, including squamous cell carcinoma (SCC), seborrheic keratosis (SK), and xanthelasma. We hope to be able to conduct more extensive experiments on ZLet-large to further explore the potential of the teacher-student model in the analysis of eyelid tumors.

## Figures and Tables

**Figure 1 jpm-13-00089-f001:**
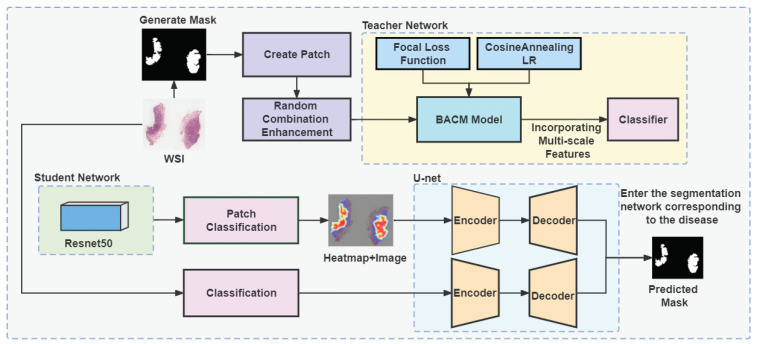
General flow-chart: the data were augmented using random combinatorial data processing; we proposed Mac-ResNet and used knowledge distillation to streamline the network. In addition, three segmentation networks were trained to learn the knowledge of three diseases and input to the corresponding class of segmentation networks to achieve the diagnosis of diseases as well as fast localization of lesion regions.

**Figure 2 jpm-13-00089-f002:**
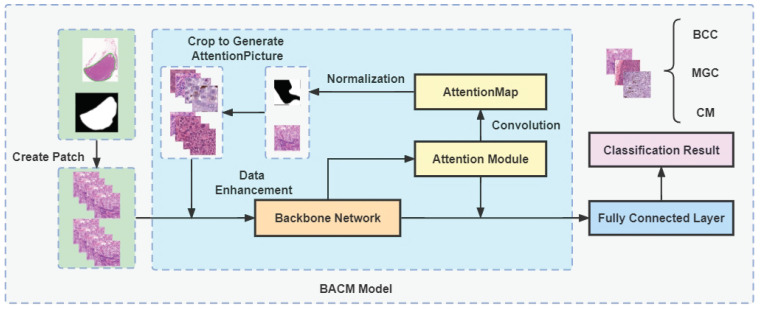
Details of BACM Model: This Network is referenced from the fine-grained classification model WSDAN and modified on its basis. The backbone network migrates the trained network parameters of the Imagenet dataset as the initial values of the network parameters.

**Figure 3 jpm-13-00089-f003:**
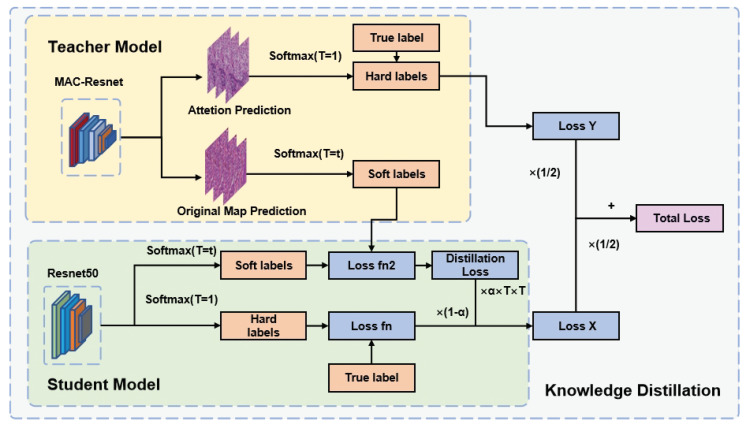
Details of Knowledge Distillation: MAC-ResNet is used as the teacher network to guide the training of student network ResNet, and the simplified network can also achieve better classification effects in the ZLet dataset.

**Figure 4 jpm-13-00089-f004:**
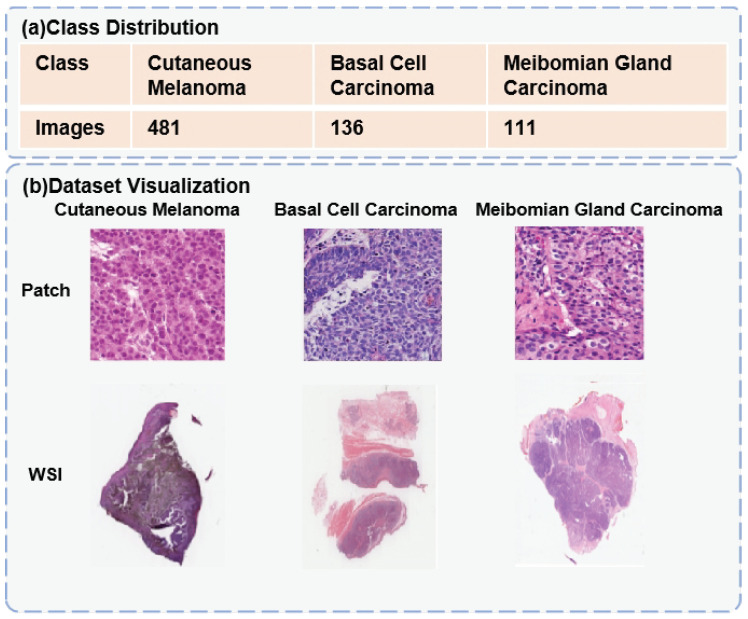
Details of Dataset: (**a**) denotes the class distribution and the number of images in the ZLet dataset, and (**b**) represents the data visualization of each type of eyelid tumor.

**Figure 5 jpm-13-00089-f005:**
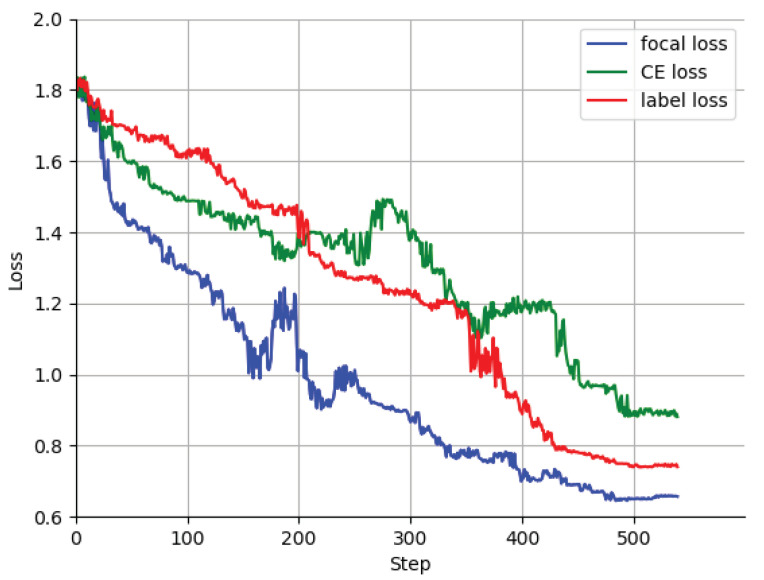
Comparative experimental results of the loss function. Due to the imbalance in classes, focal loss outcompetes other losses during the training process.

**Figure 6 jpm-13-00089-f006:**
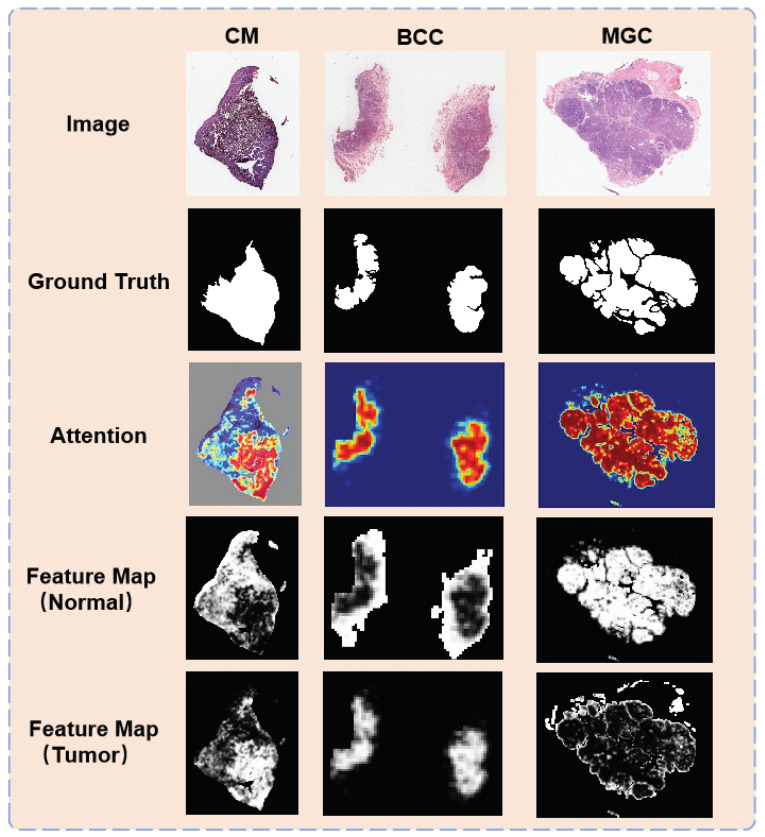
Visualization of results: The segmentation results can provide the doctor with aid in diagnosing what kind of tumor the pathology image contains and where the tumor is located.

**Table 1 jpm-13-00089-t001:** Validation set results of comparative experiments using the location of the DARes-block module.

DARes-Block Usage Location	ACC	Spec	Recall	0-ACC	1-ACC	2-ACC	3-ACC
layer2	0.8020	0.8244	0.7630	0.8672	0.7082	0.7306	0.8807
layer3	0.8110	0.8207	0.7800	0.8774	0.7150	0.7370	0.8954
layer4	0.8172	**0.8369**	0.7714	0.8858	0.7271	0.7493	0.8860
layer2 + layer3	0.8065	0.8243	0.7697	0.8695	0.7231	0.7462	0.8876
layer2 + layer4	**0.8307**	0.847	**0.8030**	**0.8873**	0.7456	**0.7785**	**0.8930**
layer3 + layer4	0.8261	0.8260	0.7963	0.8689	**0.7590**	0.78	0.8965
layer2 + layer3 + layer4	0.8187	0.8207	0.7815	0.8595	0.7476	0.7764	0.8911

**Table 2 jpm-13-00089-t002:** Validation set results for comparison tests using modified input modules.

Whether to Modify the Input	ACC	Spec	Recall	0-ACC	1-ACC	2-ACC	3-ACC
NO	0.8307	0.8470	0.8030	0.8873	0.7456	0.7785	0.8930
YES	**0.8321**	**0.8739**	**0.8140**	**0.8901**	**0.7489**	**0.7857**	**0.9035**

**Table 3 jpm-13-00089-t003:** Comparative experimental results of SPP-block.

	ACC	Spec	Recall	0-ACC	1-ACC	2-ACC	3-ACC
Without SPP-block	0.8321	0.8739	0.8140	0.8901	**0.7489**	0.7857	0.9035
With SPP-block	**0.8389**	**0.8792**	**0.8260**	**0.9135**	0.7407	**0.7914**	**0.9100**

**Table 4 jpm-13-00089-t004:** Comparative experimental results of the loss function.

Loss Function	ACC	Spec	Recall	0-ACC	1-ACC	2-ACC	3-ACC
Cross Entropy	0.8646	0.8768	0.8522	0.9209	0.7858	0.8260	0.9257
Labelsmoothing	0.8704	0.8752	**0.8803**	**0.9200**	0.7892	0.8370	**0.9354**
Focal loss	**0.8857**	**0.8835**	0.8704	0.8945	**0.8620**	**0.8710**	0.9153

**Table 5 jpm-13-00089-t005:** Comparison table of experimental results of WSAC-ResNet combination using optimized loss function and changing learning rate.

Loss	lr	ACC	Spec	Recall	0-ACC	1-ACC	2-ACC	3-ACC
CE	0.0001	0.8646	0.8768	0.8522	0.9209	0.7858	0.8260	0.9257
Focal loss	0.0001	0.8857	0.8835	0.8704	0.8945	0.8620	0.8710	0.9153
CE	CosineAnnealingLR	0.8805	0.8876	0.8692	**0.9310**	0.8176	0.8547	**0.9187**
Focal loss	CosineAnnealingLR	**0.9023**	**0.8992**	**0.9015**	0.9162	**0.8820**	**0.8937**	0.9175

**Table 6 jpm-13-00089-t006:** Overall average classification results.

Eyelid Tumor	Sensitivity	Specificity	Accuracy
BCC	0.8046	0.9862	0.9688
MGC	0.7688	0.9589	0.9467
CM	0.8889	0.9113	0.9089

**Table 7 jpm-13-00089-t007:** Overall average segmentation results.

Eyelid Tumor	IOU	Dice
BCC	0.7277	0.8349
MGC	0.6806	0.8050
CM	0.7329	0.8307

## Data Availability

Not applicable.

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
