# Peer review of "MAC-ResNet: Knowledge Distillation Based Lightweight Multiscale-Attention-Crop-ResNet for Eyelid Tumors Detection and Classification"

_jpm, 2022, doi:10.3390/jpm13010089_

Round 1
Reviewer 1 Report
This is an article entitled “MAC-Resnet: knowledge distillation based lightweight Multiscale-Attention-Crop-Resnet for ocular tumors detection and classification (jpm-2085050)” which evaluates the efficacy of deep learning techniques to investigate the pathological images of ocular tumors and propose the Multiscale-Attention-Crop-Resnet (MAC-Resnet) network model to achieve the automatic classification of three malignant tumors and the automatic localization of WSI lesion regions using U-Net.
Abstract
- Good.
Introduction
- Please re-abbreviate the terms in introduction section. It is not sufficient to just abbreviate in the abstract.
- MGC is not a common tumor of the eye.
- CM is not a common tumor of the eye either.
- Clinically, it is usually easy to discriminate MGC, BCC and CM. Most of our assumptions are correct prior to pathologic results.
Methods
- Ok.
Experiment and Results
- Ok.
Discussion
- Please write discussion. As there is only the below paragraph;
“Authors should discuss the results and how they can be interpreted from the perspective
of previous studies and of the working hypotheses. The findings and their implications
should be discussed in the broadest context possible. Future research directions may also
be highlighted.”
Conclusions
- Ok.
References
- Ok.
Author Response
Dear Reviewer:
The modification is attached in the PDF.
Thank you for your thorough review and for taking the time to provide detailed feedback on our work. We greatly appreciate the insights and suggestions that you have provided, and we will carefully consider them as we revise the manuscript. Your feedback is invaluable to us, and we are grateful for the opportunity to improve our work through the review process.
We have modified the manuscript and the detail attached in the PDF.
Thank you and best regards,
Xingru Huang

Reviewer 2 Report
The report is attached.

Author Response

(The authors gave the same response as above.)

Reviewer 3 Report
The authors proposed a deep neural network for classification and segmentation of histopathological finding in eyelid skin tumors. The subject of current study can be of interest for the readers; however, I think it needs some scientific revision before proceeding to publication. My comments and questions are as follows:
1. Despite the content of the introduction section, the dataset of the paper consists of eyelid tumors. Therefore, using the term "ocular tumor" is not accurate for the title of current study. The authors should discuss the eyelid tumors in the introduction.
2. Considering the previous comment, fundus photography is not helpful for diagnosis of mentioned tumors'
3. In figure 6, there areas of overlap between feature maps of the model for the normal and tumor regions, how do you justify these findings?
4. It seems that the discussion section is missing. The authors can compare their results with similar deep learning studies in classification of histopathology sections of skin tumors
Author Response

(The authors gave the same response as above.)

Round 2
Reviewer 1 Report
Asked revisions are made.